# A Universal Electronically Controllable Memelement Emulator Based on VDCC with Variable Configuration

**Predrag B. Petrović** 

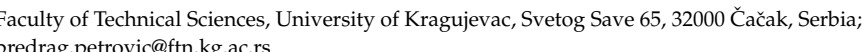

Faculty of Technical Sciences, University of Kragujevac, Svetog Save 65, 32000 Čačak, Serbia; predrag.petrovic@ftn.kg.ac.rs

**Abstract:** In this paper, a universal fractional order memelement (FOME) emulator is proposed based on the use of a voltage differentiating current conveyor (VDCC) as active block. The emulation circuit was implemented without an analog voltage multiplier and with only one type of grounded passive element—capacitors. Specially designed switching networks allow controlling the type of memelement and the emulator mode—floating or/and grounded, electronically controlled (by changing the bias voltage of the VDCC) FOMEs. The proposed emulator was theoretically analyzed, and the influence of possible non-idealities and parasitic effects was also been analyzed to reduce the undesirable effects by selecting the passive circuit elements. The proposed designs are very simple compared to most of the designs available in the literature and can operate in a wide frequency range (up to 50 MHz) and also satisfy the non-volatility test. All realized memelements can be used in incremental and decremental modes as well as in inverse configuration. The performance of the circuit was verified by HSPICE simulations using 0.18 μm TSMC process parameters and ±0.9 V power supply. The proposal is also supported by experimental results with off-the-shelf components (LM13700 and one AD844) in order to confirm the proposed solution's workability.

**Keywords:** universal FOME emulator; VDCC; grounded passive components; soft and hard switching; simulation



## 1. Introduction

The growing need for intelligent technical solutions is placing increasingly complex demands on both research and the development of complex systems that require the coordinated interaction of subsystems of different physical natures. Resistors, capacitors, and inductors are typically used as devices that model nonlinear phenomena within various physical processes. These three elements represent the fundamental relationships between the quantities current, electrostatic field, and magnetic field. Their use to describe reality is a necessity because the constituent relationships of these elements express the fundamental laws that govern reality. This trinity was extended in 1971 by the memristor [1] and in 1980 by other higher order elements (HOEs) [2,3]. The concept of memristor was also extended to meminductors and memcapacitors [4]. A memristor is a two-terminal passive circuit element that provides a non-linear relationship between the charge $q(t)$ and flux $\varphi(t)$ as in Equation (1). This relation completes the missing value among the four basic electrical magnitudes.

$$M(q) = \frac{d\varphi(q)}{dq} \tag{1}$$

In order for an electrical element providing a non-linear relationship between the flux and the charge to be referred to as a memristor, it is necessary to obtain the pinched current $i(t)$–voltage $v(t)$ hysteresis loop when a bipolar sinusoidal voltage source is applied to the element, as well as to decrease the lobe area within this loop, as the frequency of the voltage source increases. After a certain frequency, which is considered infinite, the current voltage graph becomes linear. Recently, the definition of memristive systems has been extended

to include memcapacitor and meminductor elements [4]. For the defining relations of these elements, two new electrical magnitudes are defined as $\sigma(t) = \int_{-\infty}^{t} \varphi(\tau)d\tau$ and $\rho(t) = \int_{-\infty}^{t} q(\tau)d\tau$. Similarly to the memristor element, non-linear relationships provided by these two elements are given in Equation (2).

$$
\begin{aligned}
C(\varphi) &= \frac{d\sigma(\varphi)}{d\varphi} \\
L(q) &= \frac{d\rho(q)}{dq}
\end{aligned}
\tag{2}
$$

The characteristic hysteresis loops are observed in the charge *q(t)*–voltage *v(t)* loops for the memcapacitor elements, and in the flux *φ(t)*–current *i(t)* loops for the meminductor elements [4]. Based on the type of controlling quantity, the memcapacitance/inductance and inverse memcapacitance/inductance can also be defined, as depicted for the memristor [1]. The sign of the obtained relationships (1) and (2) (time-varying part of the obtained dependence), + or −, defines the incremental or decremental mode of operation, i.e., when bringing a positive pulse of the voltage/current at the input terminal, the equivalent memresistance (memcapacitance/meminductance) increases or decrease with time, reflecting the non-volatile nature of memelements. In general, they represent a class of devices with two terminals whose inductance or capacitance is determined by the internal state of the system. Due to these properties, these elements are widely used in analog memories, adaptive filters, relaxation oscillators, neuromorphic circuits, biomedical applications, chaotic signal generators, low-power computing, programmable analog circuits, resistive random access memories (RRAM), and many others [5].

Due to the unavailability of any physical solid structure, memelements (especially meminductor and memcapacitor) have become a popular research area, especially in the last two decades. In the development of such elements, emulator circuits must mimic all of the fingerprints of ideal memelements that are well described by theoretical postulates [1–5] and must be compatible with CMOS technology, thereby becoming suitable for practical realization and verification. By the above exposure, the emulation circuits should be able to ensure control over their AC and DC performance parameters, including their operating frequency range. Since the current mode (CM) active blocks have low power consumption and a wide range of working frequencies and are minimally parasitic, they often become the base for developing new emulator solutions, due to the fact that standard operational amplifiers (OA) demand more DC power and have a lower operating frequency with a higher parasitic impact than the CM circuits. Additionally, the possible use of a multiplier increases the complexity of the circuit and reduces its operating frequency. A special effort is the development of fully functional universal emulators that are completely flexible in terms of modes of work (grounded/floating, incremental/decremental, inverse, soft or hard switching) and electronic controls. There are many solutions in the literature related to individual memelements, and a much smaller number on universal emulators, which are the topic of this paper. Due to the extensiveness of this subject and limitations of space, this paper will be mainly focused on universal emulators and solutions that have appeared in the last few years.

Over the last decade, various types of universal memelement emulators have been designed and proposed by using different active and passive elements, with most of them being able to emulate two of the three memelements. For example, the circuit proposed in [6] uses a mutator to emulate memristor and meminductance, while the design described in [7] is based on a gyrator. The memristor was the base element for the emulator proposed in [6,8]. The papers [5,9–12] describe a compact charge-controlled emulator consisting of off-the-shelf devices, based on the usage of second-generation current conveyors (CCII) and analog multipliers (AM). A very simple circuit of a memcapacitor and meminductor emulator that uses only one active block, namely, a voltage differencing current conveyor (VDCC), memristor, and capacitor, was proposed in [8]. In [13,14], a new mutator making use of a varactor diode with variable capacitance was proposed to achieve the transformation among different memelements. The study [15] presented a compact memelement

emulator circuit that realizes the behavior of meminductor and memristor based on voltage differencing transconductance amplifier (VDTA), while an emulator circuit [16] was based on a modified VDCC (MVDCC) and operational transconductance amplifier (OTA), which are CMOS-implemented electronically tunable active building blocks (ABBs). A charge-controlled memelement emulator using a VDCC and an OTA with grounded passive elements was described in [17], while a compact emulator structure that realizes the behavior of a floating meminductor and memristor employing a single voltage differencing inverted buffered amplifier (VDIBA) and a dual-output OTA was proposed in [18]. In [6], a current backward transconductance amplifier (CBTA)-based universal mutator circuit was proposed for the realization of memcapacitor and meminductor elements. Many of these universal emulators need a large silicon area and are not suitable for analog integrated circuit implementations. In the development of the circuit proposed here, the idea was to overcome this limitation by using a minimum number of active and passive components.

The proposed emulator circuits are based on the usage of the VDCC as a combination of OTA and a modified current conveyor, minimizing the number of floating elements inherent in many CCII applications [19]. The VDCC facilitates the realization of differential and dual input mode circuits. During the past two decades, it has been a popular building block in several standard analog circuits, such as oscillators, filters, and wave-shaping circuits, as well as other, non-conventional circuit concepts such as passive elements, simulators and memristors. The VDCC provides electronically tunable transconductance gain in addition to transferring voltage in its relevant terminal.

The main initial idea from which the author started is based on the desire to offer a completely flexible structure that can provide a platform for the emulation of all types of memelements that are theoretically well-defined. The possibilities offered by the proposed configuration include: grounded/floating, incremental/decremental, direct/inverse modes of operation, soft/hard switching (with the help of switches); this cannot be found in any of the previously described solutions and structures used to emulate memelements. Practically, seven completely new and original configurations have been proposed based on VDCC as an active emulation block. Depending on the specific purpose and needs of the future user, the accompanying switching structure becomes extremely simple and easy to implement, and consequently, the size of the circuit does not become too large for a future implementation in the form of an integrated circuit. Use of only one type of active block facilitates pairing and ensures better performance in practical implementation, reducing the influence of the present nonlinearities. This especially applies to the unique implementation of floating memcapacitance with a variable switching mechanism. A deeper theoretical analysis of individual configurations can be easily carried out based on already performed theoretical analyses and conclusions, and it can be the subject of some future works. The objective of this paper is to provide the circuit implementation of all three memelements and their simulation and experimental validation. The proposed memelement design can be used to explore real-world application without the need for a mutator.

The proposed configuration demonstrates the deployment of grounded capacitors and electronic controlled resistors, enabling easy monolithic integration (grounded capacitors facilitate the IC realization and compensate the stray capacitances). The proposed universal memelement emulator possesses the following benefits: it employs only two active elements—in the realization of memristors, they demand only one VDCC; the necessary passive elements (variable number of capacitors and electronically realized resistor) are grounded; it requires no external voltage multiplication circuitry (usage of one or more multipliers make these configurations bulky in size; it overcomes some of the limitations that exist in earlier reported universal memelement emulators [5,9–12]); it provides electronic controllability features (by a bias voltage of the VDCC); it does not need any type of component/parameter matching condition; it offers a maximum operating frequency of up to 50 MHz—confirmed through simulation; it does not require mutation through an external memristor to realize memcapacitor/meminductor elements. The proposed emulation circuit offers the possibility of adjusting the charge or flux value of the emulator

circuit by changing the transconductance gain of the VDCC, thereby realizing the function of compact charge/flux-controlled simulator. In the proposed floating memcapacitance mode, the circuit offers the possibility of choosing the switching mechanism—soft or hard, by modifying the value of the capacitance used, or the frequency of the current excitation signal. This type of emulator with hard switching behavior can be used in the application of spiking and bursting neuron circuits because it exhibits two states: high and low memcapacitance. The obtained simulation and experimental results are fully compliant with the relevant theoretical assumptions.

## 2. Proposed Configuration

The proposed configuration of the universal memelement emulator circuit is shown in Figure 1.

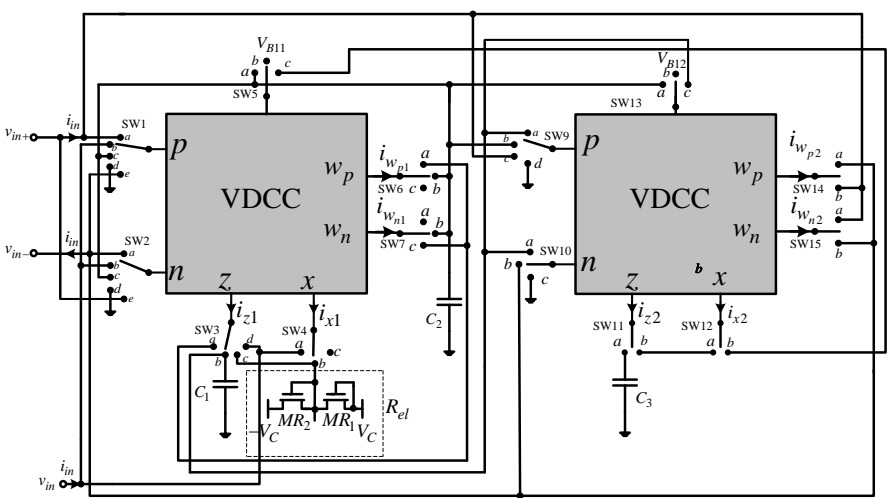

**Figure 1.** Universal memelement emulator based on VDCC.

The relatively demanding switching network that is shown in Figure 1 is the result of the idea from which the author started in the development of the universal emulator. Namely, the proposed circuit offers the possibility of both incremental and decremental modes, provides an emulation of all of the memelements described so far: the memristor, meminductor and memcapacitor. In addition, electronic control is enabled, as well as variable configuration: floating/grounded—a grounded emulator is always less applicable in circuit applications compared to a floating element. In addition, the proposed configuration also provides the possibility to work in the inverse (reverse) mode, i.e., the possibility that the generated features, instead of going through quadrants 1 and 3 (the functional dependency describing the emulated element), go through quadrants 2 and 4. The proposed switching structure can be easily realized with the use of standard programmable components, offering complete flexibility in the operation of the proposed circuit.

The terminal characteristics of VDCC can be defined by the hybrid matrix as given by Equation (3):

$$
\begin{bmatrix} i_n \\ i_p \\ i_z \\ v_x \\ i_{w_p} \\ i_{w_n} \end{bmatrix} =
\begin{bmatrix}
0 & 0 & 0 & 0 \\
0 & 0 & 0 & 0 \\
\beta g_m & -\beta g_m & 0 & 0 \\
0 & 0 & \gamma & 0 \\
0 & 0 & 0 & \alpha_p \\
0 & 0 & 0 & -\alpha_n
\end{bmatrix}
\begin{bmatrix} v_p \\ v_n \\ v_z \\ i_x \end{bmatrix}
\tag{3}
$$

As we can see in (3), the quantities of terminals $p$, $n$, and $z$ follow the function of a transconductance amplifier, while other ports ($x$, $wp$ and $wn$) including the output $z$ of the first stage are connected to the internal current conveyor stage. These two stages

can be realized by the CMOS implementation shown in Figure 2, which consists of only 24 MOS transistors.

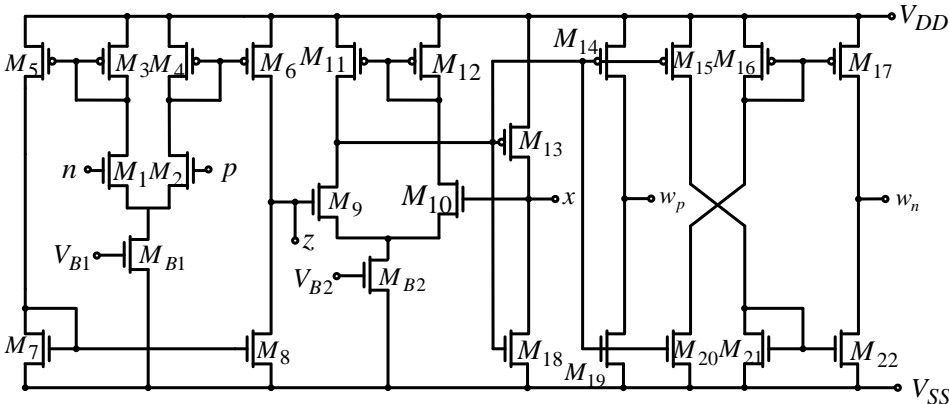

**Figure 2.** CMOS realization of VDCC.

All terminals of VDCC exhibit high impedance, except the *x* terminal. Here, $\beta$ defines the tracking error of the OTA stage of the VDCC (ideally $\beta = 1$), $\gamma$ represents the non-ideal voltage gain between *z* and *x* terminal (ideally $\gamma = 1$), $\alpha_p$ and $\alpha_n$ denotes the non-ideal current gain between *x*, $w_p$ and $w_n$ terminals (ideally $\alpha_p = \alpha_n = 1$), respectively. The transconductance gain of the VDCC can be computed as follows:

$$g_m = k(V_{B1} - V_{Tn} - V_{SS}) \tag{4}$$

where:

$$k = B\mu_n C_{ox}\sqrt{\frac{1}{2}\left(\frac{W}{L}\right)_{MB1}\left(\frac{W}{L}\right)_{MN}} \tag{5}$$

The $(W/L)_{MN}$ is defined as $(W/L)_{M1} = (W/L)_{M2}$, *W* and *L* are width and length of transistors, and *B* is denoted as the current mirroring ratio between $M_{3,4}$ and $M_{5,6}$ transistors. In addition, $\mu_n$ and $C_{OX}$ are electron mobility, an oxide gate capacitance per unit area of $MB_1$ transistor, respectively.

The equivalent resistance $R_{el}$ (realized with two p-MOS transistors) between the chosen node of VDCC and ground in Figure 1 can be tuned by a control voltage $V_c$, being defined by:

$$R_{el} = \frac{1}{2\mu_p C_{ox}(V_c - V_{Tn})}\left(\frac{1}{W/L}\right)_{MR} \tag{6}$$

where $(W/L)_{MR}$ is defined as $(W/L)_{MR1} = (W/L)_{MR2}$, while $V_c$ is the control voltage of equivalent resistor. Instead of the proposed grounded capacitances in Figure 1, we can use the MOS capacitance when the simulator circuits are operated in the high-frequency region, which brings additional benefits from the integrated circuit.

### 2.1. Charge-Controlled Grounded Memristor—Case A

When the switches are set in the following manner: SW3a (in position a)—SW4a—SW5a—SW6a—SW7b (or SW6b—SW7c—incremental or decremental function)—SW9d—SW10c, while switches SW1 and SW2 change their positions between positions b and d (direct or inverse function), Figure 3a, we can establish the following relationships:

$$v_{x1} = v_{in}; i_{x1} = -i_{in} \Rightarrow i_{w_{p1}} = -i_{w_{n1}} = -i_{in} = -i_{z1}$$
$$v_{w_{n1}} = \frac{1}{C_2}\int_0^t i_{in}(t)dt = \frac{1}{C_2}q(t) = v_{B11}(=)v_{w_{p1}} \Rightarrow g_{m1} = k(\pm v_{B11} - V_{Tn} - V_{SS}) \tag{7}$$
$$i_{z1} = i_{in} = \pm g_{m1}v_{in} = \pm k(\pm v_{B11} - V_{Tn} - V_{SS})v_{in}$$

where $q(t)$ is charge of the input current signal. The (=) in the above relation indicates the possibility to change the mode of operation: incremental or decremental, i.e., whether the bias voltage is determined as a voltage on one or the other terminal of the VDCC. It follows that:

$$R_M^{-1} = \pm k \left( \pm \frac{1}{C_2} q(t) - V_{Tn} - V_{SS} \right) \tag{8}$$

**Figure 3.** (**a**) Charge-controlled grounded memristor. (**b**) Flux-controlled grounded memristor.

Practically, the same functional dependence can be obtained if the input voltage and current are connected to port $w_n$, the port $x$ is connected to capacitance $C_1$ and at the same time with bias voltage $V_{B11}$, the port $z$ is shortly connected with port $w_p$, while ports $p$ and $n$ change positions between the input voltages and ground. In this configuration, we obtain:

$$
\begin{aligned}
&v_{p1}(v_{n1}) = v_{in}; i_{w_{n1}} = -i_{in} \Rightarrow i_{w_{p1}} = -i_{w_{n1}} = i_{in} = i_x = -i_{z1} \\
&v_{x1} = \frac{1}{C_1} \int_0^t i_{in}(t) dt = \frac{1}{C_1} q(t) = v_{B11} \Rightarrow g_{m1} = k(v_{B11} - V_{Tn} - V_{SS}) \\
&i_{z1} = \pm g_{m1} v_{in} = \pm k(v_{B11} - V_{Tn} - V_{SS}) v_{in} = -i_{in}
\end{aligned} \tag{9}
$$

Based on the above relation, we conclude that equivalent memristance possesses the same value as was defined with relation (8); however, in the above situation, the influence of parasitic impedances (which exist on all of the VDCC ports) will be different—the analysis of their impacts will be presented in the next section of the paper.

### 2.2. Flux-Controlled Grounded Memristor with Electronic Control—Case B

For the switches positioned as follows: SW3d—SW4b—SW5a—SW6b—SW7a (or SW6c—SW7b—incremental or decremental function)—SW9d—SW10c, while switches SW1 and SW2 change their position between b and d (direct or inverse function) (Figure 3b):

$$
\begin{aligned}
&v_{z1} = v_{in}; i_{x1} = \frac{v_{in}}{R_{el}} \Rightarrow i_{w_{p1}} = -i_{w_{n1}} = \frac{v_{in}}{R_{el}} \\
&v_{w_{p1}} = \frac{1}{C_2} \int_0^t i_{x1} dt = \frac{1}{R_{el}C_2} \varphi(t) = v_{B11}(=) v_{w_{n1}} \Rightarrow g_{m1} = k(\pm v_{B11} - V_{Tn} - V_{SS}) \\
&i_{z1} = -i_{in} = \pm g_{m1} v_{in} = \pm k(\pm v_{B11} - V_{Tn} - V_{SS}) v_{in}
\end{aligned} \tag{10}
$$

where $\varphi(t)$ is flux of the input voltage signal. We can conclude that:

$$R_M^{-1} = \pm k \left( \pm \frac{1}{R_{el}C_2} \varphi(t) - V_{Tn} - V_{SS} \right) \tag{11}$$

For the realization of these two (three) memristor types described above, we need only one active block—VDCC and a grounded capacitance and/or an electronically realized resistor, which makes these realizations more attractive for practical implementation than some other solutions [20,21].

### 2.3. Electronically Flux-Controlled Floating/Grounded Memristor—Case C

For the switches set as: SW1a—SW2a (or SW1e—SW2e—incremental or decremental function)—SW3b—SW4c—SW5b—SW6c—SW7a—SW13c—SW11b—SW12a—SW9c—SW10b, and switches SW14 and SW15 that change positions between levels a and b—direct or inverse function (Figure 4a):

$$
\begin{aligned}
i_{z1} &= \pm g_{m1}(v_{in+} - v_{in-}) = \pm g_{m1}v_{in} \\
&\Rightarrow \quad v_{z1} = \pm \frac{1}{C_1}\int_0^t g_{m1}v_{in}(t)dt = \pm \frac{g_{m1}}{C_1}\varphi(t) = v_{B12}(t) \\
v_{p2} &= v_{in+}; \quad v_{n2} = v_{in-} \\
&\Rightarrow \quad i_{z2} = -i_{x2} = g_{m2}(v_{in+} - v_{in-}) = k(v_{B12} - V_{Tn} - V_{SS})(v_{in+} - v_{in-}) \\
i_{x2} &= i_{w_{p2}} = -i_{w_{n2}} = \pm i_{in} \\
i_{in} &= \pm k\left(\pm \frac{g_{m1}}{C_1}\varphi(t) - V_{Tn} - V_{SS}\right)(v_{in+} - v_{in-})
\end{aligned}
\tag{12}
$$

We can conclude that:

$$
R_M^{-1} = \pm k\left(\pm \frac{g_{m1}}{C_1}\varphi(t) - V_{Tn} - V_{SS}\right)
\tag{13}
$$

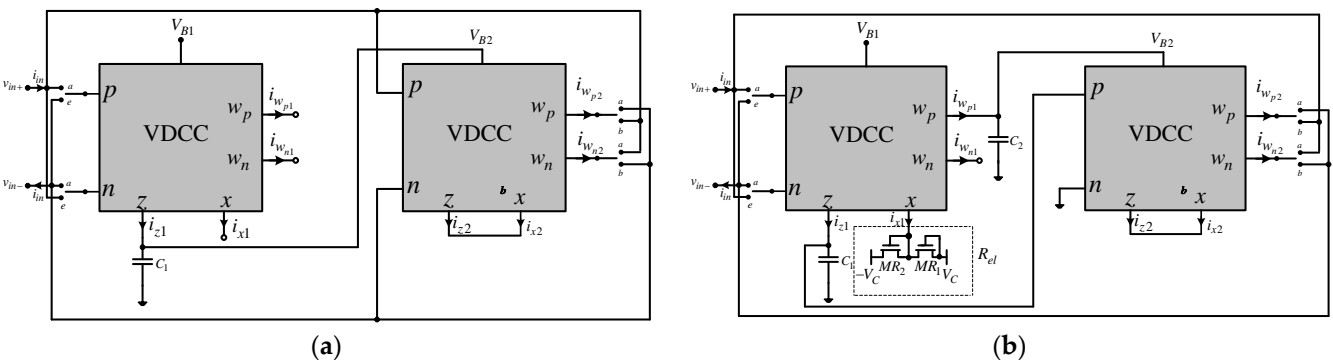

(**a**)             (**b**)

**Figure 4.** (**a**) Flux-controlled frequency floating/grounded memristor. (**b**) Floating/grounded electronically controlled meminductor.

This configuration, as can be seen from Figure 4a, does not require an output current-conveyer (second) stage in the first VDCC active block, so that the number of the required MOS transistors can be minimized when implemented in the form of an integrated circuit.

### 2.4. Floating/Grounded Electronically Controlled Meminductor—Case D

When the switches are positioned as follows: SW1a—SW2a—(or SW1e—SW2e—incremental or decremental function)—SW3b—SW4b—SW5b—SW6b—SW7a—SW9a—SW10c—SW13a—SW11b—SW12a, while switches SW14 and SW15 change their position between stages a and b—we can define the direct or inverse operation of the proposed mem-inductor-emulator circuits (Figure 4b):

$$
\begin{aligned}
i_{z1} &= \pm g_{m1}(v_{in+} - v_{in-}) = \pm g_{m1}v_{in} \quad \Rightarrow \\
&\Rightarrow \quad v_{z1} = \pm \frac{1}{C_1}\int_0^t g_{m1}v_{in}(t)dt = \pm \frac{g_{m1}}{C_1}\varphi(t) \\
i_{x1} &= \frac{v_{z1}}{R_{el}} = \frac{g_{m1}}{R_{el}C_1}\varphi(t) \Rightarrow i_{w_{p1}} = i_{x1} \quad \Rightarrow \\
&\Rightarrow \quad v_{w_{p1}} = v_{B12} = \frac{1}{C_2}\int_0^t i_{w_{p1}}(t)dt = \pm \frac{g_{m1}}{R_{el}C_1C_2}\rho(t) \\
i_{z2} &= -i_{x2} = g_{m2}v_{z1} = k(v_{B12} - V_{Tn} - V_{SS})\frac{g_{m1}}{C_1}\varphi(t) \\
i_{x2} &= i_{w_{p2}} = -i_{w_{n2}} = \pm i_{in} \\
i_{in} &= \pm k\left(\pm \frac{g_{m1}}{R_{el}C_1C_2}\rho(t) - V_{Tn} - V_{SS}\right)\frac{g_{m1}}{C_1}\varphi(t),
\end{aligned}
\tag{14}
$$

where $\rho(t)$ is time integral of flux. Based on the above derivation, it follows that:

$$L_M^{-1} = \pm \frac{kg_{m1}}{C_1}\left(\pm\frac{g_{m1}}{R_{el}C_1C_2}\rho(t) - V_{Tn} - V_{SS}\right) \tag{15}$$

The realized inverse meminductance is much less dependent on the technological parameters of the active block used than the implementation described in [22], with a slight increase in the number of used MOS transistors. In this way, it is easier to control the area of the pinched hysteresis loop (obtaining a smooth pinched hysteresis loop at high frequencies without any changes in $f$ and amplitude of the input signal), as well as reduce the impact of present non-idealities.

### 2.5. Electronically Controlled Grounded Memcapacitor—Case E

For the switches SW3c—SW4a—SW5c—SW6c—SW7b—(or SW6b—SW7a incremental or decremental function)—SW9b—SW10c—SW13b—SW11a—SW12b, and the switches SW1 and SW2 changing the positions between b and d (direct or inverse function), Figure 5, we can obtain:

$$
\begin{aligned}
&v_{x1} = v_{in}; i_{x1} = -i_{in} \quad \Rightarrow \quad i_{w_{p1}} = -i_{w_{n1}} = -i_{in} \\
&v_{w_{n1}} = \frac{1}{C_2}\int_0^t i_{in}(t)dt = \frac{1}{C_2}q(t) = v_{p2}(=)v_{w_{p1}} \quad \Rightarrow \quad i_{z2} = \pm\frac{g_{m2}}{C_2}q(t) \\
&v_{z2} = v_{x2} = v_{B11} = \frac{1}{C_3}\int_0^t i_{z2}(t)dt = \pm\frac{g_{m2}}{C_2C_3}\sigma(t); g_{m1} = k(v_{B11} - V_{Tn} - V_{SS}) \\
&i_{z1} = \pm g_{m1}v_{w_{n1}} = \pm g_{m1}\frac{1}{C_2}q(t) \quad \Rightarrow \quad v_{z1} = v_{in} = i_{z1}R_{el} \\
&v_{in} = \pm k\left(\pm\frac{g_{m2}}{C_2C_3}\sigma(t) - V_{Tn} - V_{SS}\right)\frac{R_{el}}{C_2}q(t)
\end{aligned}
\tag{16}
$$

where $\sigma(t)$ is time integral of charge. It follows that:

$$C_M^{-1} = \pm\frac{kR_{el}}{C_2}\left(\pm\frac{g_{m2}}{C_2C_3}\sigma(t) - V_{Tn} - V_{SS}\right) \tag{17}$$

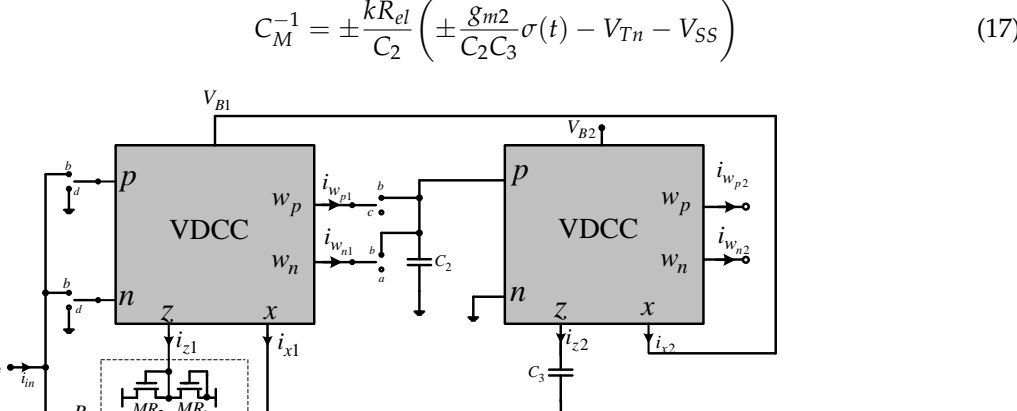

**Figure 5.** Grounded memcapacitor emulator.

### 2.6. Floating/Grounded Memcapacitor with Hard/Soft Switching Mechanism—Case F

The emulation of this memelement type is the most complex, and such a compact realization as proposed herein is unknown in the literature thus far. The realization described in [23] is practically based on two active elements defined as a hybrid structure, with very complex requirements that must be complied with by the bias voltages. In addition, the influence of the nonlinearity present is much more pronounced than with the proposed solution here, which supports a significantly lower frequency range in which the circuit may yield the described performances. In addition to the already presented, the theoretical setting described in [23] is unsustainable, as a transconductance parameter used in the relation (18) of the paper [23] was defined based on the value derived in (19),

thus obtaining a differential dependence of the second order with functional coefficients. Practically in one iteration, there is a transconductance parameter defined through the bias voltage, and already in the next, it is defined through the generated voltage at the capacitor, which is completely wrong.

For the circuits in Figure 6, we can conclude, based on functional relations between VDCC ports, that:

$$
\begin{aligned}
i_{W_{p1}} &= -i_{W_{n1}} = i_{x1} = i_{in} \Rightarrow v_{x1} = \frac{1}{C_1}\int_0^t i_{in}(t)dt = \frac{1}{C_1}q(t) = v_{p2}(t)(=)v_{n2}(t) \\
i_{z2} &= g_{m2}v_{p2}(v_{n2}) \quad \Rightarrow \\
&\Rightarrow \quad v_{z2} = \frac{1}{C_2}\int_0^t i_{z2}(t)dt = \pm\frac{g_{m2}}{C_1C_2}\int_0^t q(t)dt = \pm\frac{g_{m2}}{C_1C_2}\sigma(t) = v_{B11}(t) \\
i_{z1} &= \pm g_{m1}(v_{in+} - v_{in-}) = \pm k(v_{B11}(t) - V_{Tn} - V_{SS})(v_{in+} - v_{in-}) \\
v_{z1} &= R_{el}i_{z1} = v_{x1} = \frac{1}{C_1}q(t) \quad \Rightarrow \\
&\Rightarrow \quad \pm kR_{el}(v_{B11}(t) - V_{Tn} - V_{SS})(v_{in+} - v_{in-}) = \frac{1}{C_1}q(t) \quad \Rightarrow \\
&\Rightarrow \quad (v_{in+} - v_{in-}) = \pm\frac{1}{kR_{el}\left(\pm\frac{g_{m2}}{C_1C_2}\sigma(t) - V_{Tn} - V_{SS}\right)C_1}q(t)
\end{aligned}
\tag{18}
$$

It follows that the equivalent memcapacitance is:

$$
C_M^{-1} = \pm\frac{1}{kR_{el}\left(\pm\frac{g_{m2}}{C_1C_2}\sigma(t) - V_{Tn} - V_{SS}\right)C_1}
\tag{19}
$$

The choice of the terminal of VDCC to which the input voltage of the proposed emulation circuit is connected—position of switches SW1 and SW2—determines the shape of the characteristic of the realized memcapacitor, i.e., whether the pinched hysteresis loop passes through quadrants 1 and 3 or 2 and 4 of the *v-q* characteristic—direct or inverting memcapacitance (the negative memcapacitance). In addition, the emulator circuits proposed in Figure 3 offer the possibility of soft or hard switching. The *v-q* loop changes slowly from the high memcapacitance state to low memcapacitance state in the smooth switching mechanism. The memcapacitor that exhibits a smooth switching mechanism can be used in nonlinear, sensitive memory, chaotic and neuromorphic circuit applications. In a hard switching mechanism, the memcapacitor behavior switches suddenly from high to low state. The memcapacitor with hard switching mechanism exhibits two states: high and low memcapacitance, such as transistors, and can be treated as two terminal transistors.

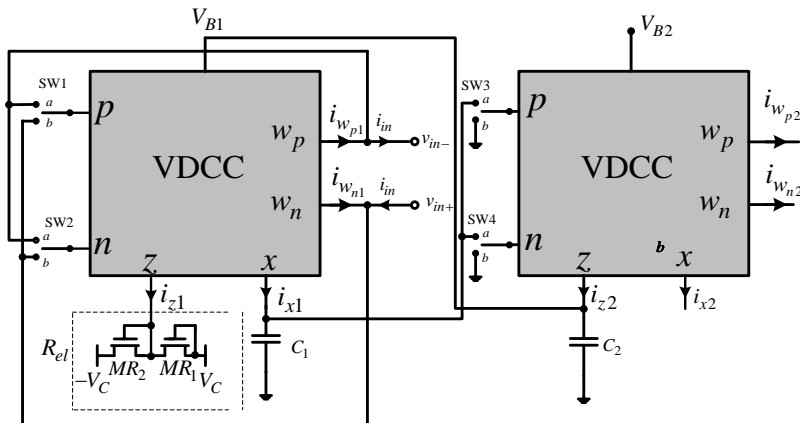

**Figure 6.** Floating/grounded memcapacitor emulator.

The type of switching depends on the position of the breakpoints on the pinched hysteresis loop, which are determined by the moment when the charge on the proposed emulator circuits reaches the maximum value [24], e.g., the moment when the input current

signal reaches zero crossing. Assuming that the input current is defined as $I_m\cos\omega t$ and the value of the voltage converges to zero [24], the obtained transient characteristic approaches the $q$-axis. By such adjustment, it is possible to obtain the hard switching characteristic. On this basis, the value of the input voltage at the moment when the direction of movement of the operating point on the transient characteristic changes is defined as

$$
\begin{aligned}
v(t_1) &= \mp \frac{R_{el}I_m}{k[V_{SS}+V_{Tn}]\omega C_1} \to 0 \\
\frac{\omega k C_1}{R_{el}I_m}&[V_{SS}+V_{Tn}] \gg 1
\end{aligned}
\tag{20}
$$

From the obtained condition, it can be seen that changing the value of the frequency of the current excitation signal or the size of the capacitor leads to a change in the switching characteristics—wide of the pinched hysteresis loop (far from the pinched hysteresis loop)—while the choice of the input voltage terminal can also realize the inverting memcapacitance characteristic. The emulator circuits with hard switching characteristics can be used in the application of a spiking and bursting neuron circuit. In contrast to [24], the configuration described here allows the choice between incremental and decremental operation, depending on the setting, by selecting the polarity of the generated bias voltage—$v_{B11}(t)$. However, it should be noted that even without this option, it is possible to switch the circuit to inverse operation due to the functional dependence between the input voltage and the charge and provide an incremental mode of operation (in direct mode, the memcapacitor emulator operates in decremental mode), since the value of the memcapacitance then increases from the initial negative value—Equation (19).

## 3. Non-Ideal and Parasitic Analysis

In real practical conditions, the VDCC possesses non-ideal gains—tracking errors, as was described in the text below the relation (1), for which reason the obtained relations (8), (11), (13), (15), (17), and (19) for memelements can be reformulated in the following way:

$$
R_M^{-1} = \pm \frac{k\beta_1}{\alpha_{p1}}\left( \pm \frac{\alpha_{n1}}{C_2}q(t) - V_{Tn} - V_{SS} \right)
\tag{21}
$$

$$
R_M^{-1} = \pm k\beta_1\left( \pm \frac{\alpha_{p1}\gamma_1}{R_{el}C_2}\varphi(t) - V_{Tn} - V_{SS} \right)
\tag{22}
$$

$$
R_M^{-1} = \pm k\gamma_2\beta_2\left( \pm \frac{\beta_1 g_{m1}}{C_1}\varphi(t) - V_{Tn} - V_{SS} \right)
\tag{23}
$$

$$
L_M^{-1} = \pm \frac{k\beta_2\alpha_{p2}\gamma_1\beta_1 g_{m1}}{C_1}\left( \pm \frac{\alpha_{p1}\gamma_1\beta_1 g_{m1}}{R_{el}C_1C_2}\rho(t) - V_{Tn} - V_{SS} \right)
\tag{24}
$$

$$
C_M^{-1} = \pm \frac{k\beta_1\alpha_{n1}R_{el}}{\gamma_1 C_2}\left( \pm \frac{\alpha_{n1}\gamma_2\beta_2 g_{m2}}{C_2C_3}\sigma(t) - V_{Tn} - V_{SS} \right)
\tag{25}
$$

$$
C_M^{-1} = \pm \frac{\gamma_1}{k\beta_1 R_{el}\left( \pm \frac{\alpha_{p1}\beta_2 g_{m2}}{C_1C_2}\sigma(t) - V_{Tn} - V_{SS} \right)C_1}
\tag{26}
$$

For the modified proposal in case A (memristor realized with different switch configuration, described in text below relation (6)), the equivalent memristance in a non-ideal situation will be:

$$
R_M^{-1} = \pm k\beta_1\left( \frac{1}{\alpha_{p1}C_2}q(t) - V_{Tn} - V_{SS} \right)
\tag{27}
$$

The non-ideal gains—tracking errors—are in the range of 0.9 to 1, with an ideal value of 1. Generally, these tracking factors remain constant and frequency-independent within low to medium frequency ranges. From all of the above relations, it was concluded that the proposed universal emulator exhibits the magnitudes of all of the sensitivity values—normalized passive and active sensitivities are equal to or less than unity in

magnitude. Based on this, we can conclude that the proposed circuits offer low passive and active sensitivities.

The parasitic resistances $R_p$, $R_n$, $R_z$, $R_{wp}$ and $R_{wn}$ and the parasitic capacitances $C_p$, $C_n$, $C_z$, $C_{wp}$ and $C_{wn}$ [25,26] appear in parallel at the corresponding terminals $p$, $n$, $z$, $w_p$ and $w_n$ (in an ideal VDCC, all of these parasitic resistances are approximately equal to infinity, while all parasitic capacitances are approximately equal to zero—in the form of a shunt $R$-$C$ network at all of the terminals with external circuit elements), while at port $x$, parasitic resistance $R_x$ and parasitic inductance $L_x$ appear in series (in ideal VDCC, $R_x$ and $L_x$ are approximately equal to zero). If we take into consideration the influence of the parasitic existence of parasitic port impendences, we come to a position where it is possible to analyze the operation of the proposed universal memelement emulator in a situation when it operates in a high-frequency environment (at high frequencies, parasitic port resistances can be neglected) and define the equivalent functional dependencies as:

$$R_M^{-1} = \pm \frac{k\beta_1}{\alpha_{p1}} \left( \pm \frac{\alpha_{n1}}{(C_2 + C_{w_{n1}})} q(t) - V_{Tn} - V_{SS} \right) \tag{28}$$

$$R_M^{-1} = \pm k\beta_1 \left( \pm \frac{\alpha_{p1}\gamma_1}{(R_{el} + R_{x1} + sL_{x1})R_{el}C_2} \varphi(t) - V_{Tn} - V_{SS} \right) \tag{29}$$

$$R_M^{-1} = \pm k\gamma_2\beta_2 \left( \pm \frac{\beta_1 g_{m1}}{(C_1 + C_{z1})} \varphi(t) - V_{Tn} - V_{SS} \right) \tag{30}$$

$$L_M^{-1} = \pm \frac{k\beta_2\alpha_{p2}\gamma_1\beta_1 g_{m1}}{(C_1 + C_{z1})} \left( \pm \frac{\alpha_{p1}\gamma_1\beta_1 g_{m1}}{(R_{el} + R_{x1} + sL_{x1})(C_1 + C_{z1})\left(C_2 + C_{w_{p1}}\right)} \rho(t) - V_{Tn} - V_{SS} \right) \tag{31}$$

$$C_M^{-1} = \pm \frac{k\beta_1\alpha_{n1}R_{el}}{\gamma_1(1 + sR_{el}C_{z1})(C_2 + C_{w_{n1}} + C_{p2})} \left( \pm \frac{\alpha_{n1}\beta_2\gamma_2 g_{m2}}{(C_2 + C_{w_{n1}} + C_{p2})(C_3 + C_{z2})} \sigma(t) - V_{Tn} - V_{SS} \right) \tag{32}$$

$$C_M^{-1} = \pm \frac{\gamma_1(1 + sR_{el}C_{z1})}{k\beta_1 R_{el} \left( \pm \frac{\alpha_{p1}\beta_2 g_{m2}}{(C_1 + C_{p2})(C_2 + C_{z2})} \sigma(t) - V_{Tn} - V_{SS} \right) C_1} \tag{33}$$

For the proposed modified configuration in case A, parasitic impendences of VDCC ports does not have an influence on the obtained value of equivalent memresistance (relation (27)), for which reason this design is very attractive for practical implementation.

On the base of the obtained relations (28)–(33), in order to preserve the performance of the proposed universal emulation circuits, the external capacitors $C_1$, $C_2$ and $C_3$ must be chosen in such a way that they are several times higher than the parasitic capacitances of the ports to which they are connected. Owing to this setting, the parasitic capacitance effects can be absorbed at working frequencies. The resistance of the equivalent electronically controlled resistor must be greater than the series resistance of the $x$ port. In order to reduce the impact of the parasitic resistances, the values of $C_1$, $C_2$ and $C_3$ must be selected in such a way that the equivalent impedance of these capacitors is several times smaller than the parasitic resistances of the VDCC ports.

Additionally, the transconductance of the OTA cell in the VDCC is a frequency-dependent parameter-determining factor, and its bandwidth limitation can be described by a single pole model. For this reason, the transconductance gain and the value of parameter $k$ of the first stage of the VDCC can be defined as follows:

$$g_m = g_{m0}\frac{\omega_g}{(s + \omega_g)}; \quad k = k_0\frac{\omega_k}{(s + \omega_k)} \tag{34}$$

where $g_{m0}$ is the transconductance gain and $k_0$ is the gain factor of the OTA cell at zero (low) frequency, while $\omega_g = 1/\tau_g$ and $\omega_k = 1/\tau_k$ are the corresponding pole frequencies (the $\tau_g$, and $\tau_k$ are delays corresponding to pole frequencies, respectively). Simply replacing values

for parameters $g_m$ and $k$ in the above Equations (28)–(33), more complete images will be obtained about the frequency characteristics of the realized emulators and their dependence on present non-idealities. It is completely clear that the useful operating frequency range of the emulator in Figure 1 can be defined as $\omega \ll \min(\omega_g, \omega_k)$. In general, the values of these pole frequencies in (34) will depend on the practical implementation of the VDCC. The bandwidth of the VDCC can be improved by inserting a compensation resistor $R$, one voltage buffer and additional MOS transistor pair, as proposed in [27] in the VDCC CMOS structure shown in Figure 2. With this modification, the transconductance gain is changed—the new value of this parameter becomes defined as $g_m = g_{m0}/(1 + g_{m0}R)$. In this way, the bandwidth of the OTA cell (first stage of the VDCC) can be changed because it depends on $g_m$.

## 4. Simulation and Experimental Results

The VDCC-based universal memelement emulator was simulated using the HSPICE software, and the various operating performance values were analyzed using level 49 TSMC CMOS 0.18 μm process model parameters. The DC supply and $V_{B2}$ voltages of the circuit were chosen as $\pm 0.9$ V and 0 V, respectively. The bulk terminals of the PMOS and NMOS transistors were connected to their sources terminals and the most negative voltage point ($V_{SS}$), respectively. The capacitor value and the aspect ratios of the VDCC were chosen as in [17,26], and $M_{B1}$, $M_{B2}$, $MR_1$, and $MR_2$ were selected as 3.6 μm/1.8 μm, 3.06 μm/0.72 μm, 60 μm/2 μm, and 60 μm/2 μm, respectively. An electronic resistor that is implemented by only two PMOS transistors can be tuned by the control voltage, where its control voltage of 0.65 V resulted in equivalent resistance of $R_{el} = 1.47$ kΩ. The power consumption of the proposed VDCC (Figure 2) was 0.869 mW. The layout of the VDCC (in 0.18 μm technology) along with electronically controlled resistor and switch was shown in [28] excluding the capacitor, and occupied a 42.2 μm × 27.5 μm chip area.

The frequency response of the current transfer gain, the tracking error of the transconductance gain, and voltage transfer gain for VDCC are shown in Figure 7a,b. The simulation results show that transfer gains retained a constant of up to 100 MHz.

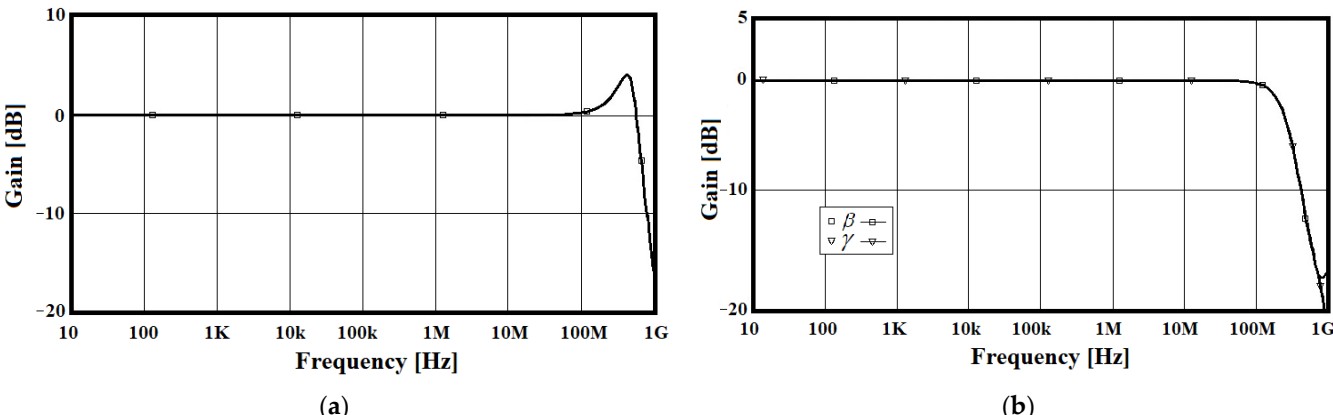

**Figure 7.** (**a**) Frequency response for the current transfer gain ($\alpha$). (**b**) Frequency response for the tracking error of the transconductance gain of the VDCC ($\beta$) and the voltage transfer gain ($\gamma$).

Important DC and AC parameters of the used VDCC (Figure 2) are summarized in Table 1 (with obtained values of port parasitic impedances). The min/max value of the bias voltage was $-0.4/-0.1$ V; in this range, the VDCC retained the characteristics defined here.

**Table 1.** DC and AC parameters of VDCC.

| Parameter | Value for $V_{B1} = -0.28$ V |
|---|---|
| $p$ and $n$ dc resistance, $R_p = R_n$ | 4. 38 TΩ |
| $z$ output dc resistance $R_z$ | 229.43 kΩ |
| $w_p$ and $w_n$ output dc resistance | $R_{wp}$ = 186.66 kΩ<br>$R_{wn}$ = 175.28 kΩ |
| $p$ and $n$ input capacitance $C_p = C_n$ | 0.034 pF |
| $w_p$ and $w_n$ output capacitance | $C_{wp}$ = 0.0105 pF<br>$C_{wn}$ = 0.022 pF |
| $z$ output capacitance $C_z$ | 0.025 pF |
| $x$ output dc resistance $R_x$ | 43 Ω |
| $z$ output inductance $L_x$ | 1.28 µH |
| transconductance of OTA stage $g_m$ | 282 µS |
| corner frequency $\omega_g$, $\omega_k$ | 5.6 rad/s |

Figure 8a shows the simulation results at an excitation current input signal frequency of 1 MHz and an amplitude of 250 µA for different values of grounded capacitance $C_2$ in case A, when the proposed design simulated grounded charge-controlled memristance. In Figure 8b, the simulation check was performed for different frequencies in case B, at a constant value of the capacitance used and the amplitude of the excitation voltage signal of 0.2 V. With the increase in the working frequency, it was concluded that the area covered by the lobes on the *i-v* plane was becoming reduced, which is a well-known characteristic of memristive elements. The VDCC as an active device was also used in the implementation of memristive emulators described in [21,28], but the configurations proposed here (cases A and B) are far superior in terms of performance in virtually all aspects: extended frequency range, simpler implementation with fewer passive components, lower impact of existing non-idealities. The maximum operating frequency of the proposed emulator circuits was 50 MHz (Figure 8c) for the proposed configuration in case C, as a consequence of realistically achievable capacitance values in the integrated technique, directly resulting from the size of the capacitor used ($C_1$), since it becomes the order of magnitude of parasitic capacitances that exist on VDCC ports (Table 1).

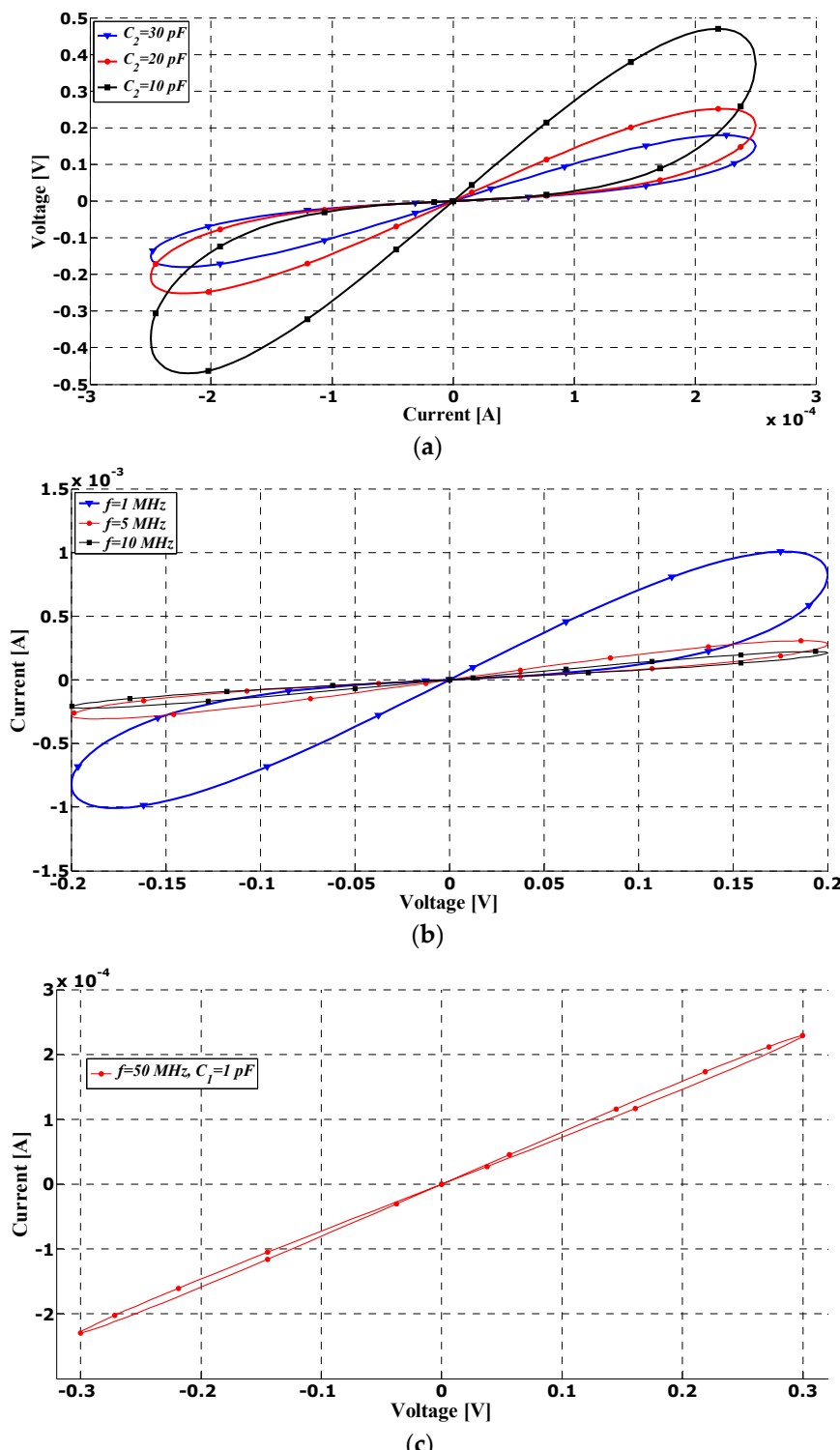

**Figure 8.** Pinched hysteresis loops generated by the proposed memristor emulators (**a**) for different capacitances, *f* = 1 MHz—case A; (**b**) for different frequencies of the input voltage signal and grounded capacitance $C_2$ = 100 pF—case B; (**c**) for frequency of 50 MHz and amplitude $V_m$ = 300 mV—case C.

The meminductor emulator—case D of the proposed universal emulator, as the other solutions—possesses the electronically adjustable feature because the flux value of the circuit can be adjusted by changing the transconductance parameter of the VDCC. For the different bias voltages, the input current–flux relationships of the proposed meminductor emulator are shown in Figure 9a. To show the transient characteristic of the proposed

meminductor emulator, the simulation was performed for a sinusoidal input voltage signal of 10 kHz frequency and 200 mV amplitude (Figure 9b). The current did not appear as a sinusoidal waveform, since the inductance value changed over time. As the frequency of the voltage signal applied to the input of the circuit increased, the phase difference between the input current and the flux value of the meminductor decreased. The proposed configuration offers the possibility of much better control of the emulated meminductance value, as well as a wider frequency range with fewer used passive components compared to circuits proposed in [29]. Based on the functional dependence obtained for the emulated meminductance value (13), the increase in the frequency of the input voltage signal will come from decreasing the dynamic range of flux due to the decrease in the value of the first term in relation (13), which has an inverse dependence on the frequency of the processed voltage signal.

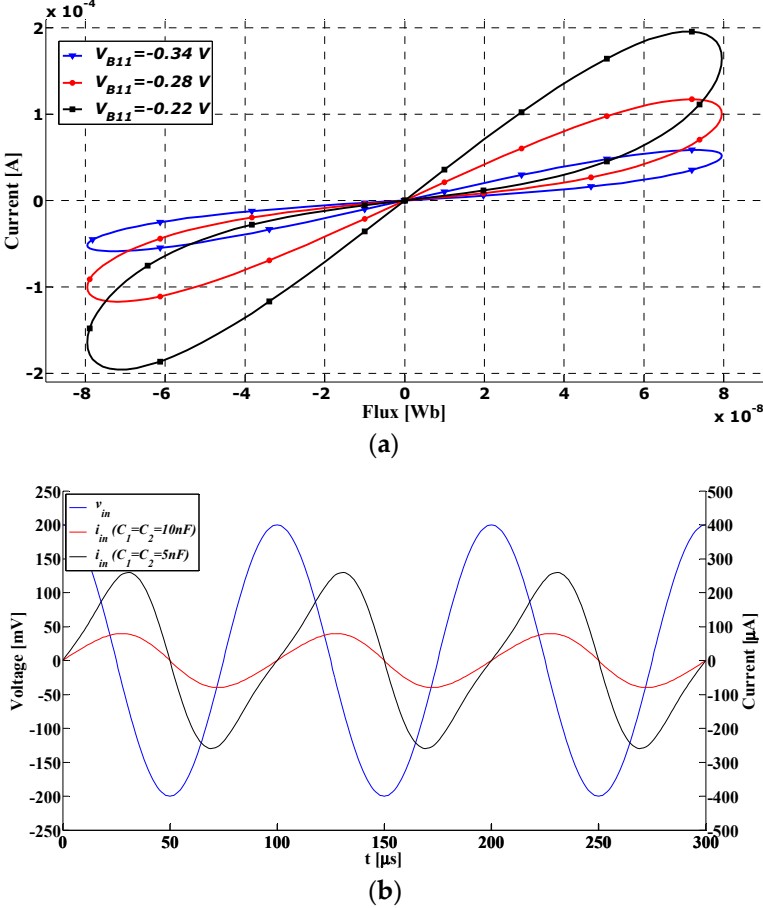

**Figure 9.** Transient responses of the proposed emulator circuits: (**a**) the input current–flux relationships for the floating/grounded meminductance emulator—case D, for different bias voltages of VDCC ($C_1 = C_2 = 50$ pF, $f = 1$ MHz, $V_m = 0.5$ V, $V_{B11} = -0.28$ V); (**b**) time-domain response of the proposed meminductor emulator.

To show the transient characteristic of the proposed structures in case E—the time domain response of the proposed memcapacitor emulator—(Figure 10a), the simulation was performed for a sinusoidal input current signal amplitude of 100 µA and frequency of 10 kHz. As we can see from Figure 7a, the generated voltage signal does not have a sinusoidal waveform, because the value of memcapacitance changes relative to time, while with the increasing frequency of the input current signal, the phase difference between the input voltage signal and the charge value of the memcapacitance decreases. The temperature performance of the circuit described in case E was analyzed on the basis of the simulation performed with three different temperatures (Figure 10b). Based on

the presented characteristics, the circuits retained the projected properties over a wide temperature range ($C_2 = C_3 = 50$ pF, $f = 1$ MHz, $I_m = 0.25$ mA, $V_{B12} = -0.28$ V). It is also important to note that with falling temperature levels, the current flow in the memcapacitor was enhanced.

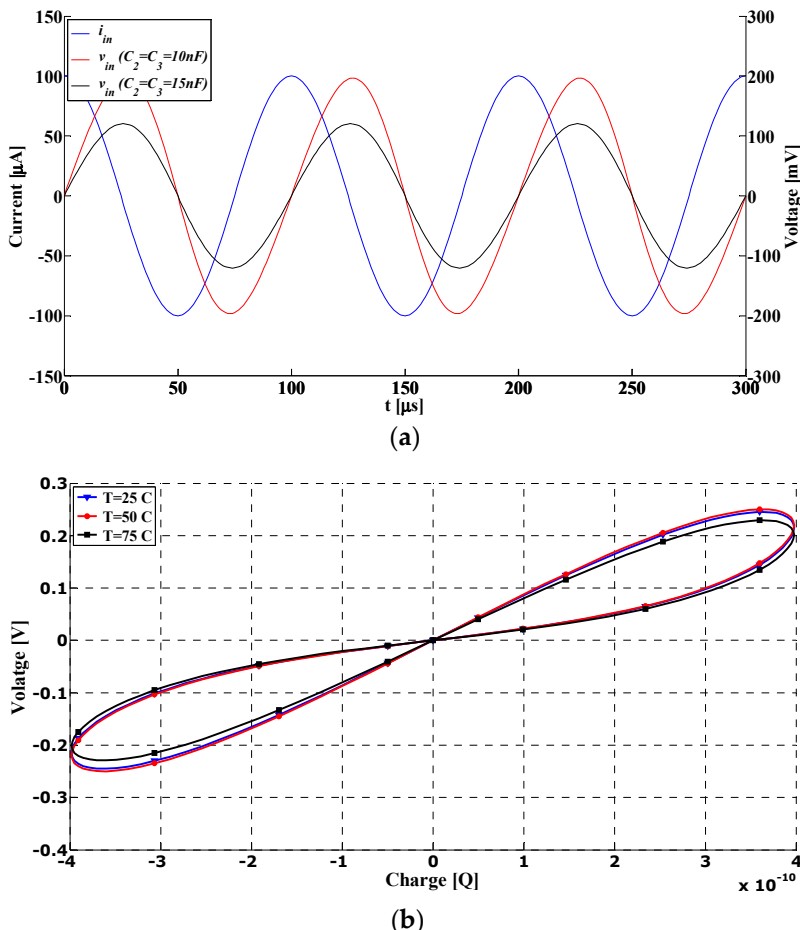

**Figure 10.** (**a**) Time-domain response of the proposed memcapacitor emulator in case E, (**b**) hysteresis loop at various temperatures.

An input sinusoid current signal with a frequency of 10 MHz and amplitude $I_m = 200$ μA with 90° phase shift was used in the course of simulation of the configuration proposed in case F—floating memcapacitance. To demonstrate the transition from one switching mechanism to another in case F, in Figure 11a, different values of ground capacitances $C_1$ and $C_2$ were used. By increasing and decreasing the capacitor values, the pinched hysteresis curve area decreased and increased, respectively, reducing the capacitance. The bending point on the transient characteristic became closer to the $q$ axis, i.e., the value of the voltage corresponding to that point converged to 0, as predicted through theoretical analysis (relation (18)). The same effect can be obtained by changing the frequency of the input current signal because the linear time-invariant part of the emulated memcapacitance becomes dominant over the linear time-variant part (Equation (19)). As was predicted with the theoretical analysis, the memcapacitor emulator proposed in case F can operate in inverting mode. The performance check of this working regime was performed for different frequencies and amplitudes of the excitation signal, at a constant value of the capacitance used ($C_1 = C_2 = 100$ pF) (Figure 11b). On the basis of the simulation results, we can conclude that with an increase in the working frequency, the area covered by the lobes on the $q$-$v$ plane is reduced. In the case of an increase in the operating frequency, the pinched hysteresis curve area decreases, and the memcapacitor behaves like an ordinary capacitor because the variable part in Equation (17) diminishes as the frequency of operation increases.

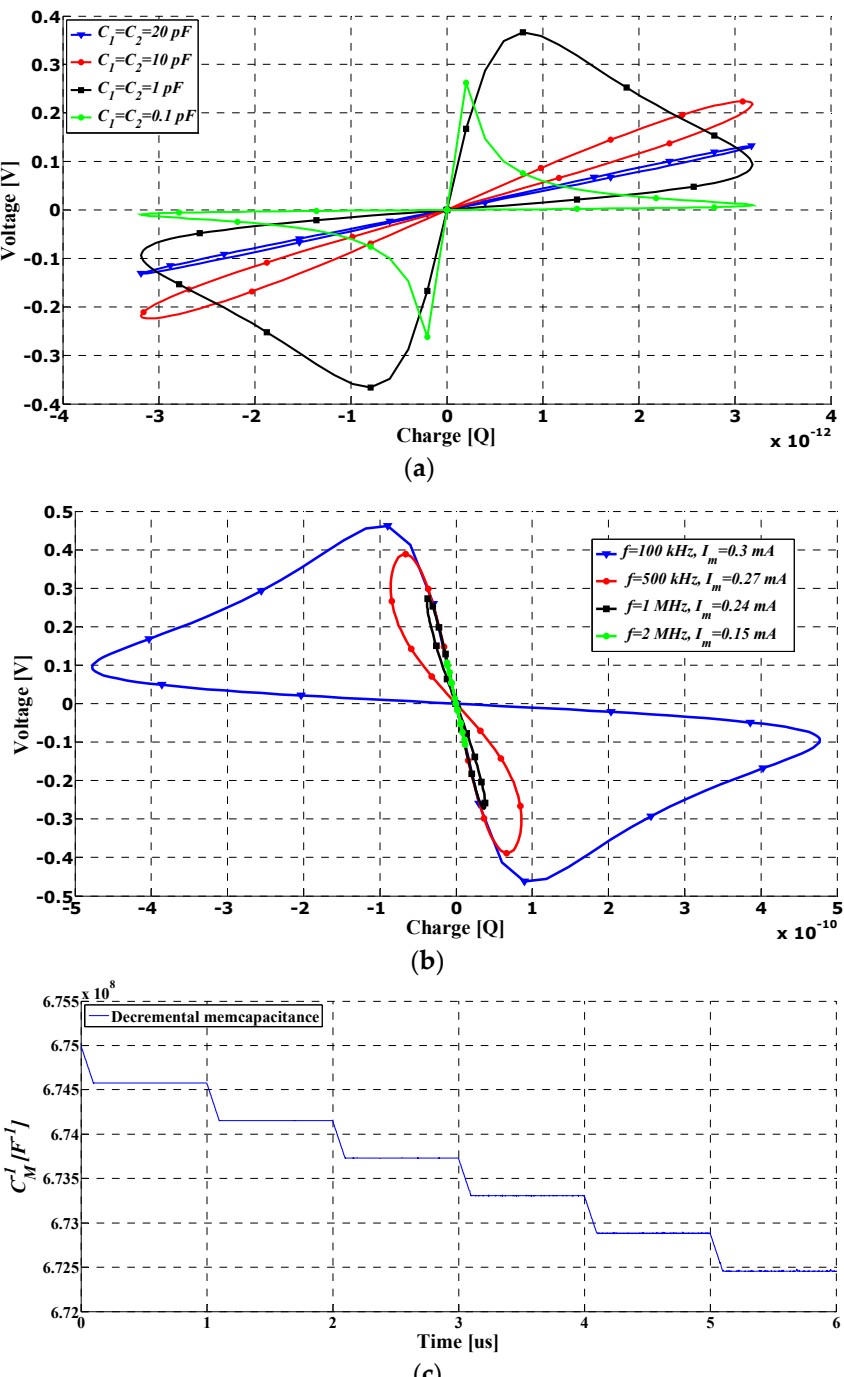

**Figure 11.** Pinched hysteresis loops generated by the charge-controlled memcapacitor emulators proposed in case F (**a**) for different capacitances, $f = 10$ MHz, $I_m = 0.2$ mA; (**b**) for different frequencies and amplitudes of the input current signal in inverting mode of operation, $C_1 = C_2 = 100$ pF; (**c**) variation of memcapacitance with time for pulse voltage of $T_{on} = 0.1$ μs for circuits proposed in case F at 1 MHz.

The non-volatile nature of the memcapacitor proposed in case F was investigated as one of the important characteristics of the mem-system. To demonstrate this feature, Figure 11c shows the variation in the emulated memcapacitance with time, having a pulse input amplitude of 0.5 mA, with a period of 1 ms and a pulse width of 0.1 ms ($C_1 = C_2 = 1$ nF, $V_{B11} = -0.28$ V). As we can see from the presented simulation results, the memcapacitance value remained constant even in the absence of a pulse signal—the proposed circuit showed

strong memory properties between pulses. Practically, the proposed configuration (as all others proposed in this paper) is suitable for the study and design of neuromorphic circuits provided with synaptic plasticity, and in particular, a long-term potentiation (LTP). Additionally, by switching the proposed emulation circuit into the inverting mode, it is possible to provide an incremental mode of operation, since then the memcapacitance value increases from the initial negative value (Equation (19)).

### 4.1. Process Variation

Process corners such as Nominal NMOS Nominal PMOS (NN), Fast NMOS Fast PMOS (FF), Slow NMOS Slow PMOS (SS), Fast NMOS Slow PMOS (FS), and Slow NMOS Fast PMOS (SF) variations are analyzed in Figure 12, hence the investigation of process variations being absolutely crucial for monolithic integration [15–18]. In this way, we came to a position to evaluate the robustness, as well as the uncertainty of the proposed realizations of the universal emulator, using the HSPICE software package to simulate variation from the manufacturing process. This study was carried out at 10 MHz with $C_1 = C_2 = C_3 = 10$ pF, $V_m = 0.2$ V and $I_m = 0.3$ mA. Based on the obtained simulation results, depending on the simulated configuration, we can conclude that the voltage/current flow in the case of FF mode is larger than in SS mode, as expected, yielding a lower voltage/current flow in the SS process corner when compared with the FF process corner.

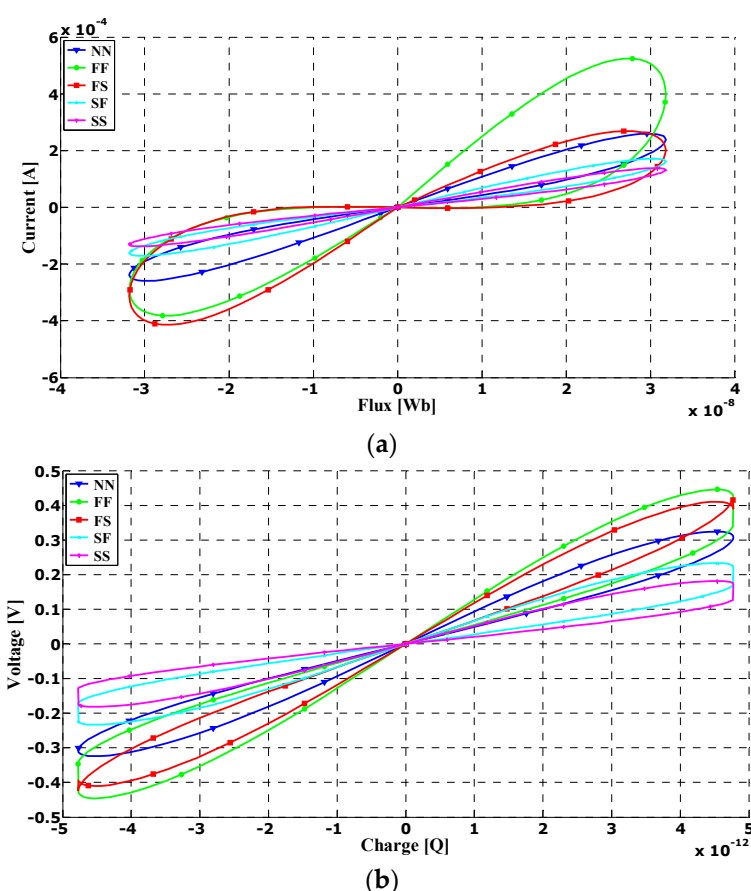

**Figure 12.** Pinched hysteresis loop variation at different process corners: (**a**) case D; (**b**) case F.

The proposed universal mememulator circuits exhibited a pinched hysteresis loop in all process corners, and no offset was observed in the characteristics. The emulator configurations proposed in cases D and F, as the most complex, were used for the analyses, in order to assess the impact of variable parameters on the achieved performance in the most direct way. This choice was certainly conditioned by the limitation of space, which was insufficient to carry out such a check for all of the described configurations. Practically,

we can conclude that the proposed design can be operated successfully in wide temperature ranges and different radical conditions. The performances of the proposed configurations in this way were practically tested when the parameters used in the simulation process differed, which included the impact of the changed dimensions of the MOS transistors used on the parameters found in the HSPICE library—some parasitic capacitances would be over- or (usually) underestimated. A much more complex and precise analysis can be carried out on this issue, using the procedure described in [30].

### 4.2. Experimental Results

To better view the realistic performances of the proposed universal emulator, since the VDCC was not a commercially available component, the circuit shown in Figure 1 was built with off-the-shelf electronic devices. The commercial IC-based implementation of the VDCC is based on the configuration proposed in [17] and demands the use of one LM13700 and one AD844 (Figure 13a). Measurements were performed using a Digilent Analog Discovery 2 board and probes, while the supply voltages were set to $\pm 5$ V. The amplitude of the input current signal varied from $I_m = 0.4$ mA to $I_m = 0.5$ mA, while in cases of experimental checks of flux-controlled emulators, the input voltage signal possessed amplitudes of $V_m = 0.4$ V and $V_m = 0.5$ V.

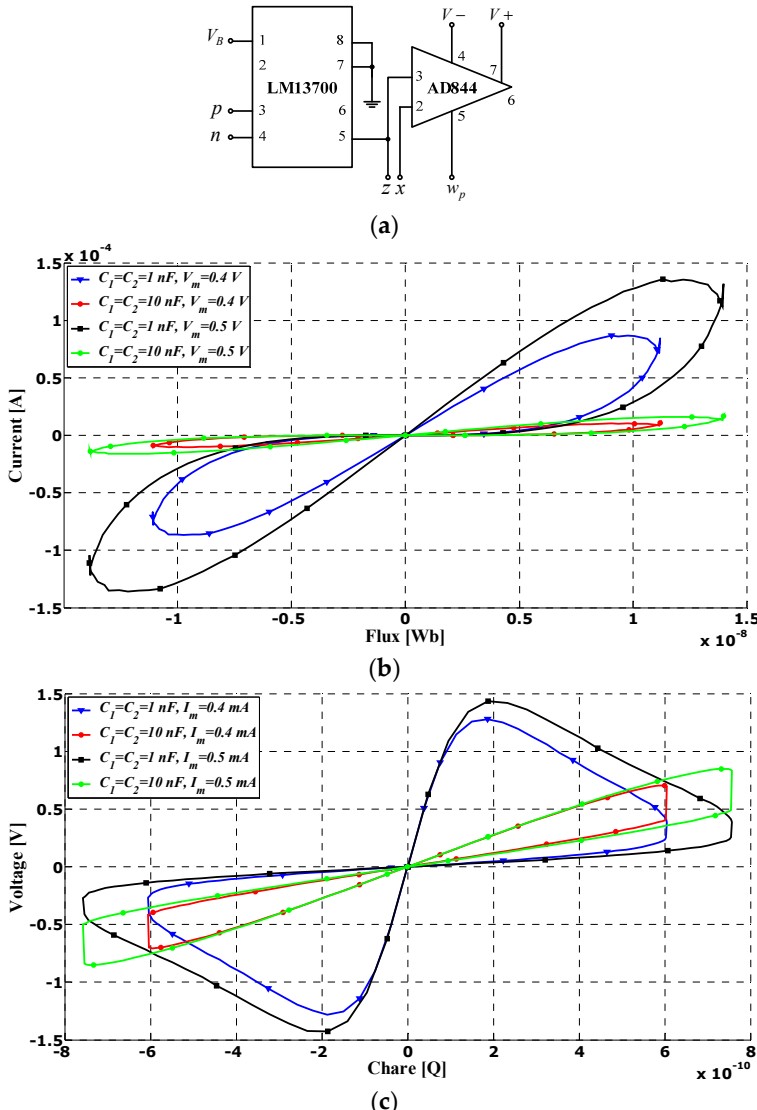

**Figure 13.** (**a**) Practical implementation of VDCC; experimental result of pinched hysteresis loop at 10 kHz obtained using commercial ICs: (**b**) case D; (**c**) case F.

The pinched hysteresis loops shown in Figure 13 were obtained for the operating frequencies of 100 kHz (as a result of the limitations of available off-shelf devices) and by scaling down the value of *C*. Such operating frequencies do not require special compensation techniques and the use of an additional number of passive elements, which was done in [12,14]. The obtained measurement results with the experimental setup were similar to those of the simulated one. It was also proved that by scaling down the capacitor value (in case F), the pinched hysteresis loop can become closer to the *q-axis*, that is, the switching mechanism changes. The $R_{el}$ value was replaced with a resistance of 100 $\Omega$, and the $g_m$ value was adjusted to 10 mS for the experiment.

Table 2 highlights the different design- and performance-related aspects of the previously reported universal memelement emulators and of the one proposed here—especially those that have emerged in recent years. The comparison was made by applying several different criteria—the number of active and passive components, electronic tunability, frequency bandwidth, mode of operation, and power supply. Based on the presented characteristics, it can be concluded that the proposed emulator circuit offers the possibility of operating at the highest operating frequency of all known circuits, with very low consumption. Thanks to the possibility that the shape of the obtained pinched hysteresis loops can be adjusted electronically, the described emulator is very attractive for applications that require compensation of parameter variation by adjusting the values of their parameters. Based on the presented parameters, the circuits whose designs were described in the paper are superior for the simplicity and compactness of their topology, bandwidth, as well as for the possibilities they offer for direct implementation in the form of an integrated circuit. The proposed design allows easy conversion from a charge-controlled to a flow-controlled counterpart and vice versa, which is possible with a simple conversion of two terminals [31] and also in an inverting configuration, which allows conversion from a nonlinear current-controlled element to a nonlinear voltage-controlled HOE and conversion from a voltage-controlled element to a current-controlled element. The above conversion capabilities make the proposed solution useful for redesigning the nonlinear properties of memristive devices, which could be marketed as solid-state devices with rigid nonlinearities [31].

**Table 2.** Comparison of existing universal memelement emulators with the proposed emulator.

| Ref. | Number of Active Comp. | Emulated Elements | Number of Grounded/Floating Passive Elements | Power Supply | Max. Operating Frequency | Type of Emulator (F/G) | Electron. Tunability | Need for External Memristor |
|---|---|---|---|---|---|---|---|---|
| [5] | 3 CCII, 1 AM 1 OA, 1 divider | MR MC ML | 5 G | ±5 V | 5 kHz | G | No | No |
| [6] | 1 CBTA | MC ML | 1 G | ±0.9 V | 300 kHz | F | Yes | Yes |
| [8] | 1 VDCC | MC ML | 1 G | ±0.9 V | 700 kHz | F | Yes | Yes |
| [9] [meas.] | 3 CCII, 1 AM _____ 5 CCII, 1 AM, 1 OP | MR MC ML | 5 G _____ 6 G, 1 F | ±10 V | 5 kHz | G _____ F | No | No |
| [10] [meas.] | 2 CCII 1 AM | MR MC | 3 G, 2 F 2 G, 2 F | ±10 V | 25 kHz | G F | No | No |
| [11] | 2 (1) CCII, 1 MOTA (OTA) | MR MC ML | 4 G (2 G) | ±1.2 V | 1 MHz | G(FMR) | No | No |
| [12] [meas.] | 4 AD844, 1 AD633, 1 μA741 | MR MC ML | 2 G, 6 F | ±15 V | 10–560 kHz 48–360 kHz 20 k–1.5 MHz | F/G | No | No |

**Table 2.** *Cont.*

| Ref. | Number of Active Comp. | Emulated Elements | Number of Grounded/Floating Passive Elements | Power Supply | Max. Operating Frequency | Type of Emulator (F/G) | Electron. Tunability | Need for External Memristor |
|---|---|---|---|---|---|---|---|---|
| [13] [meas]. | 4 AD844, 1 TL084, 1 VD | MR MC ML | 4 G, 4 F | ±10 V | 10 kHz | F | No | No |
| [14] [meas]. | 4 AD844, 1 µA741, 1 VD | MR MC ML | 1 G, 3 F (2 G,2 F) | ±15 V | 180 kHz | F | No | No |
| [15] | 2 VDTA | MR ML | 2 G | ±0.9 V | 1.5 MHz | F | Yes | No |
| [16] | 1 MVDCC, 1 OTA | MR ML | 3 G | ±0.9 V | 300 kHz | F | Yes | No |
| [17] | 1 VDCC, 1 OTA | MR MC | 3 G | ±0.9 V | 1 MHz | G | Yes | No |
| [18] | 1 VDBIA, 1 OTA, 2 MOS | MR ML | 1 G 2 G | ±1 V | 4–8 MHz 2 MHz | F | Yes | No |
| [32] | 3 MOOTA, MOS | MC MI | 3 G | ±0.9 V | 500 kHz | G | Yes | No |
| This work | 2 VDCC, 2 MOS FET | MR-A | 1 G | ±0.9 V | 2 MHz | G | No | No |
| | | MR-B | 1 G | ±0.9 V | 50 MHz | G | Yes | No |
| | | MR-C | 1 G | ±0.9 V | 50 MHz | F/G | Yes | No |
| | | ML-D | 2 G | ±0.9 V | 50 MHz | F/G | Yes | No |
| | | MC-E | 2 G | ±0.9 V | 2 MHz | G | Yes | No |
| | | MC-F | 2 G | ±0.9 V | 50 MHz | F/G | Yes | No |

F = floating; MR = memristor; G = grounded; MC = memcapacitor; AM = analog multiplier; ML = meminductor; VD = varactor diode; [meas]. = measured on real devices.

## 5. Conclusions

This paper proposes a completely new structure for a universal variable-mode memelement emulator, based on the application of a VDCC, and—depending on the configuration—memelements that are emulated, with only two grounded passive components, making it an attractive solution from the point of view of an integrated circuit. The transconductance parameter of the VDCC provides a way to adjust the pinched hysteresis loop irrespective of the frequency and amplitude values of current/voltage across emulator circuits. The effects of the VDCC non-idealities, including transfer gain errors and parasitic elements on the realized memelement emulator, were investigated, making it possible to properly select the passive circuit elements. Confirmation of the proposed concept (obtained based on theoretical assumptions and conclusions) was performed through simulation and experimental verification using off-the-shelf components. Variable analysis based on a variety of parameters such as process variation, capacitor variation, temperature variation, and frequency variation was carried out, whereby the simulation confirmed the possibility of working at frequencies of up to 50 MHz. Additionally, the realized floating/grounded memcapacitance can provide two different switching mechanisms—soft and hard—and can simulate a negative memcapacitance characteristic that makes this emulator an excellent candidate for the realization of nonlinear chaos oscillators and neuromorphic computing. For a better analysis of the performance, the proposed universal memelement emulator was compared with the existing solutions and showed better performance than the existing ones, overcoming some of the limitations that existed in the implementation of previously known solutions in terms of consumption, maximum operating frequency and number of used active blocks in the integrated implementation.

**Funding:** This research was supported by the Ministry of Education, Science and Technological Development of the Republic of Serbia, and these results are parts of the Grant No. 451-03-68/2022-14/200132 with University of Kragujevac—Faculty of Technical Sciences Čačak.

**Institutional Review Board Statement:** Not applicable.

**Informed Consent Statement:** Not applicable.

**Conflicts of Interest:** The author declares no conflict of interest.

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
