# Peer review of "A Universal Electronically Controllable Memelement Emulator Based on VDCC with Variable Configuration"

_electronics, doi:10.3390/electronics11233957_

Round 1

Reviewer 1 Report

I am truly sorry, but I do not understand the significance of this paper.

The circuit size is too large to be implemented in a real integrated circuit. If it is just for theoretical evaluation, it is easier to use a compact model in a circuit simulator. 

So, please clarify the significance of your research in Introduction.

Author Response

Reviewer # 1

Thank you for your comments which helped me to improve the presentation and quality of my paper.

Comments #1: I am truly sorry, but I do not understand the significance of this paper. The circuit size is too large to be implemented in a real integrated circuit. If it is just for theoretical evaluation, it is easier to use a compact model in a circuit simulator. So, please clarify the significance of your research in Introduction.

Author response: In the new version of the paper, the author has done everything in his power to offer a completely accurate answer to this comment. Namely, the main initial idea from which the author started is based on the desire to offer a completely flexible structure that can provide a platform for the emulation of all types of memelements that are theoretically well-defined and described in the professional literature. In doing so, the author adhered to the guidelines defined in the preparation and invitation of this special edition of the journal. The possibilities offered by the proposed configuration include: grounded/floating, incremental/decremental, direct/inverse mode of operation, soft/hard switching; this cannot be found in any of the so far described solutions and structures used to emulate memelements. This also required the complex switching structure described in the paper. Practically, seven completely new and original configurations have been proposed based on VDCC as an active emulation block. Depending on the specific purpose and needs of the future user, the switch structure becomes extremely simple and easy to implement. This becomes even clearer after the changes made by the author in the new version of the paper, where, at the suggestion of one of the reviewers, each of the configurations is accompanied with an additional image that clarifies the switching network corresponding to the given configuration. For this reason, the size of the circuit does not become too large for a future implementation in the form of an integrated circuit, which was confirmed by many referenced works that described the layout of the DVCC. The author based the solution on the use of only one type of active block, which facilitates pairing and ensures better performance in practical implementation, reducing the influence of the present nonlinearities. This especially applies to the unique implementation of floating memcapacitance with a variable switching mechanism. Practically each of the described configurations could represent a special scientific contribution - which would lead to a completely compact circuit structure for simulation - however, the author wanted to summarize all the possibilities that the proposed configuration offers in one paper. This is another confirmation of the attractiveness of the proposed emulation circuit, as it is based on an active block that has shown its good performance in several realizations that are described in the professional literature. Certainly, I take this opportunity to thank the reviewer for his sincerely useful suggestions.

Reviewer 2 Report

In the current form the paper seems to be a patchwork - neither in-depth theoretical analysis, nor feasibility study for CMOS implementation, nor the serious benchtest of laboratory model composed from of shelf devices. All this three perspectives deserve dedicated - preferably separate, but anyway more elaborated papers.

Abstract - The performance of the circuit has been verified by Pspice simulations using 0.18 μm TSMC 18 process parameters and ± 0.9 V power supply and also validated experimentally - this statement is highly inappropriate - experimental verification was done using not the manufactured circuit designed by the author, but that made from off shelf components! So the statement made is very unfair.

Moreover important figures are in wide discrepancy  - while simulation results mention 50 MHz (line 15 of the manuscript) the measurements done with undergraduate student's setup - using of which - by the way - is used as a kind of "excuse" - show the operating frequecy of 10 kHz (line 513) - i.e three and half order of magnitude lower.  Therefore data included in the table 2 comparing results with other autors and claiming "both" as MHz figures were measured are very misleading to the prospective reader! 

In general even simulation results are disputable. Author uses Ppspice simulator - while TSMC design kit (at least that avaliable via Europractice) provides user with Eldo, Spcectre and HSPICE models). No mentions about adoption of models dedicated for different tool. Furthermore since no layout tool is mentioned it seems that presented simulations are pre-layout, thus probbably not including drains/source areas/perimeters - so consequently many parasitic capacitances (not mentioning path resistances) are missing and frequency prerformance is significantly overestimated.

The text in lines 480-484 the author mixes-up the concepts of PVT corners and mismatch-mode Monte Carlo simulation.  Redirecting potential reader to [29] (line 482) as a reference for Monte Carlo (.MC syntax is quite different it PSpice and HSPICE which is used by autors of [29]!) analysis or eventually source of corner models (NDAs ?!) is somewhat weird - did the author of the manuscript want to satisfy the editor citing the paper from the same journal?

In lines 88-89 the author follows the current-conveyor "propaganda" presenting expectations about the block at the time of its conceptual introduction (late '60) as reality. This attitude has been criticized by Hanspeter Schmid already  two decades ago (AICaSProc., 35, 79–90, 2003)

Author Response

Reviewer # 2

I would like to use this opportunity to thank you for the time dedicated to my paper. The author would like to express sincere gratitude to the reviewer for contributing the remarks that have helped to improve the quality of the paper and to make it much clearer and more applicable. The new version of the paper has a much clearer layout and is easier to follow. The imprecisions that were present in the first version of the paper have now been removed, following the recommendations from the reviewer.

Comment # 1: In the current form the paper does not seem to be a patchwork - neither in-depth theoretical analysis, nor feasibility study for CMOS implementation, nor the serious benchtest of laboratory model composed from of shelf devices. All this three perspectives deserve dedicated - preferably separate, but anyway more elaborated papers.

Author response: I can fully understand this view of the respected reviewer, but the author tried to describe in the most precise and compact way all the possibilities offered by the proposed platform, based on the use of only one type of active element - VDCC. A deeper theoretical analysis of individual configurations can be easily carried out based on already performed theoretical analyses and conclusions, and it can be the subject of some future works. The author did not do this for several reasons: to avoid repeating already known approaches that have been elaborated in many published works and are well known to the professional audience, and also due to the limitation of space. The intention was to describe as many completely new and functional configurations as possible. On the other hand, CMOS implementation with all its requirements and problems that it brings with it has already been described in the works that treated that issue in the context of VDCC implementation. Through a simulation check, the author came to the values of important parameters that confirm the applicability of VDCC in the realization of the configuration offered here.

Comment # 2: Abstract - The performance of the circuit has been verified by Pspice simulations using 0.18 μm TSMC 18 process parameters and ± 0.9 V power supply and also validated experimentally - this statement is highly inappropriate - experimental verification was done using not the manufactured circuit designed by the author, but that made from off shelf components! So the statement made is very unfair.

Author response: All the statements made on this issue by the reviewer are completely true. The author, as described in the text of the paper, conducted the experimental verification based on the use of not the manufactured circuit designed by the author, but rather the one made from off-shelf components. So as to avoid ambiguity on this issue, this is precisely explained in sufficient detail in the abstract of the paper. The author uses the opportunity to once again thank the reviewer for the perceived omission made in the previous version of the paper.in the realization of the configuration offered here.

Comment # 3: Moreover important figures are in wide discrepancy - while simulation results mention 50 MHz (line 15 of the manuscript) the measurements done with undergraduate student's setup - using of which - by the way - is used as a kind of "excuse" - show the operating frequency of 10 kHz (line 513) - i.e. three and half order of magnitude lower. Therefore data included in the table 2 comparing results with other authors and claiming "both" as MHz figures were measured are very misleading to the prospective reader!

Author response: As in the case of the previously made statements, the facts presented here are completely in place. In the experimental verification of the proposed concept, based on commercially available components, the author conducted measurements at frequencies that were used in works of this type, and in the new version of the work, it was done for the operating frequency of 100 kHz. Such operating frequencies do not require special compensation techniques and the use of an additional number of passive elements, which was done in works that were used as a reference and which used only commercially available integrated components for emulation. The author plans to realize the proposed design in the form of an integrated circuit - however, it is currently technically and financially impossible to organize this either at the institution where he works or at any other institution and faculty in Serbia. If there is an opportunity to establish cooperation with one of the foreign institutions, it would speed up the solution to this problem as well. What stands as a current possibility is only the estimation of the dimensions of the layout circuit without the possibility of realizing practical measurements – the layout of such a proposed VDCC is well described in many papers. In Table 2, where the comparison was made, for the reasons listed here, the column pointed to by the reviewer was removed, so that there would be no confusion on this issue. Through accompanying comments in the text of the paper, additional clarification is provided of the actual performance of the proposed solution, as well as the circuit with which the comparison was made. In the new version, it is presented in a transparent way and explained so that there would be no additional confusion on that issue. The new version of the paper specifies whether these performances are the result of simulation tests or realistically recorded characteristics, so that on this basis, any doubts have been removed.

Comment # 4: In general even simulation results are disputable. Author uses Ppspice simulator - while TSMC design kit (at least that avaliable via Europractice) provides user with Eldo, Spcectre and HSPICE models). No mentions about adoption of models dedicated for different tool. Furthermore since no layout tool is mentioned it seems that presented simulations are pre-layout, thus probbably not including drains/source areas/perimeters - so consequently many parasitic capacitances (not mentioning path resistances) are missing and frequency prerformance is significantly overestimated.

Author response: To avoid confusion on this issue, and in full accordance with the proposal given, in the new version of the paper, it is precisely specified that the HSPICE models, level 49, were used. The author believed that this should not be emphasized in the previous version of the paper, implying that mentioning Pspice as a software platform for simulation checks will not confuse the reader, since it is clear (as it was done in the reference works) that such a simulation check implies an adequate model of the used MOS transistors. I thank the reviewer for the observed omission. The model used is very detailed, and includes drains/source areas/perimeters, so it can be considered that the obtained simulation results reliably indicate the expected performance of the proposed emulator as if it were realized in the form of an integrated circuit.

Comment # 5: The text in lines 480-484 the author mixes-up the concepts of PVT corners and mismatch-mode Monte Carlo simulation. Redirecting potential reader to [29] (line 482) as a reference for Monte Carlo (.MC syntax is quite different it PSpice and HSPICE which is used by authors of [29]!) analysis or eventually source of corner models (NDAs ?!) is somewhat weird - did the author of the manuscript want to satisfy the editor citing the paper from the same journal?

Author response: The author fully supports the objection presented here. The wording that was used confuses what was done - process corners analysis, and mismatch-mode Monte Carlo simulation. In the new version of the work, it is fully clarified, sticking to the concept that was also supported by other authors, who were involved in the design of emulator circuits. The cited paper is listed with the desire to guide the reader that a much more complex and precise analysis can be carried out on this issue, which is described in the mentioned paper, and not for the reason of satisfying the editor citing the paper from the same journal.

Comment # 6: In lines 88-89 the author follows the current-conveyor "propaganda" presenting expectations about the block at the time of its conceptual introduction (late '60) as reality. This attitude has been criticized by Hanspeter Schmid already two decades ago (AICaSProc., 35, 79–90, 2003)

Author response: At this point, I sincerely thank the reviewer for the observed fact about "propaganda" presenting the current conveyor. Indeed, in many places in the professional literature, this performance is mentioned as an important advantage of the current conveyor, which can be the subject of justified criticism. This part of the text of the paper has been completely reformulated following the comment presented.

Reviewer 3 Report

The Author proposes an universal emulator for memelements based on 2 VDCC, a grounded resistor and a large switch network needed to reconfigure the system to cover all the possible configurations.
My comments:
1) To make the paper more readable, it would be useful to add a short section defining the memelements and the different options considered (charge-based and flux-based, incremental and decremental etc).
2) CBTA in the Introduction has not been defined.
3) Eq. (2) and (3) are referred to a specific implementation, that is later shown in Fig. 2. Probably it could be better to move them after Fig. 2.
4) To make the paper more readable, it could be useful to add figures showing the connections of the switches described in the text ((e.g. small versions of Fig. 1), similarly to what has been done in Fig. 3.
5) Eq. (18) is not clear without the definition of soft/hard switching.
6) The Author could comment the compensation at the end of Sec. 3.

Author Response

Reviewer # 3

At this point, the author must thank the reviewer for his generally positive attitude towards the results they came to while working on this project and the idea. Your attitude gives me additional strength and confidence that I am on the right path. Thanks again for everything, as well as for the time you have dedicated to my work, as well as for highly useful and good-natured comments that have enabled me to significantly improve my work. The new version, at least I hope, and I deeply believe it, is much more precise in its fortifications and more comprehensible to the broader reading audience. For their part, the author has done everything within his power to respond in a more precise and clearer way to all the supplied comments. For all of this, I owe a great thank you to the reviewer.

Comment # 1: To make the paper more readable, it would be useful to add a short section defining the memelements and the different options considered (charge-based and flux-based, incremental and decremental etc).

Author response: Through the redacted, new version of the paper, the authors paid special attention to these issues, for which they thank the reviewer most sincerely. In the introductory part of the paper, additional explanations are introduced which define more precisely the emulated quantities and their functional dependence. This is explained in sufficient detail in the new version of the paper.

Comment # 2: CBTA in the Introduction has not been defined.

Author response: In the new version of the paper, this acronym for the active circuit is defined, it is the current backward transconductance amplifier - CBTA.

Comment # 3: Eq. (2) and (3) are referred to a specific implementation, that is later shown in Fig. 2. Probably it could be better to move them after Fig. 2.

Author response: As the esteemed reviewer suggested in his comment, the equations have been moved in the text, to the section after Figure 2.

Comment # 4:To make the paper more readable, it could be useful to add figures showing the connections of the switches described in the text ((e.g. small versions of Fig. 1), similarly to what has been done in Fig. 3.

Author response: Figures showing the connections of the switches described in the text are given for all the proposed configurations, although all this led to an increase in the volume - space of the work itself.

Comment # 5:Eq. (18) is not clear without the definition of soft/hard switching.

Author response: As the reviewer suggested in his comment, along with equality 18, a definition of the obtained - realized switching mechanisms is given, which makes it possible to see even more the contributions of the proposed solution and its flexibility in this matter as well.

Comment # 6:The Author could comment the compensation at the end of Sec. 3.

Author response: Through the newly introduced comments and the newly introduced reference that was used as the basis for the proposed compensation, the author hopes that there are no more ambiguities on this issue. As with all previous comments, the author sincerely thanks the reviewer for the perceived omissions and ambiguities that existed in the previous version of the paper.

Round 2

Reviewer 1 Report

Thank you for your reply. But, I am sorry but my conclusion in the last review does not change. Namely, the circuit size is too large to be implemented in a real integrated circuit. If it is just for theoretical evaluation, it is easier to use a compact model in a circuit simulator. So, I believe that this manuscript is not suitable to the academic journals such as "electronics". Some review journal or educational journals might be suitable.

Author Response

Reviewer # 1

Comments: Thank you for your reply. But, I am sorry but my conclusion in the last review does not change. Namely, the circuit size is too large to be implemented in a real integrated circuit. If it is just for theoretical evaluation, it is easier to use a compact model in a circuit simulator. So, I believe that this manuscript is not suitable to the academic journals such as "electronics". Some review journal or educational journals might be suitable.

Author response: I thank the esteemed reviewer again for the time and attention he has given to my work. I am sorry to say that the changes and comments in the latest version of the paper do not respond to the comments in the expected way.

I must disagree with the expressed opinion that the circuit is too large, because its dimensions require only one active block and a minimal number of passive, grounded components for some of the configurations, which facilitates its realization in the form of an integrated circuit and reduces parasitic effects. Compared to all previously known solutions of universal emulators in this dimension, the solution proposed here has the advantage of minimizing the number of active blocks used. In accordance with the theme of this special issue of the journal, the contribution of the work refers to completely new and original configurations, very sophisticated in their working concept, to emulator circuits superior in all parameters important for such systems. The integrated form itself, which is certainly important for further implementation, is not the focus of this type of research. Solutions based on memristors but using only three MOS transistors and passive components are known in the literature, but their performance is significantly limited, which is overcome by the concept proposed here.

The layout of the VDCC (in 0.18mm technology) along with electronically controlled resistor and switch was specified in [28] excluding the capacitor, and occupies 42.2 mm x 27.5 mm chip area.

The author has adhered to the concept and presentation style that is common in papers of this type and that he has used several times in the preparation of the paper, which has been published in various journals, the last paper having just appeared in this journal. Also, in the cases where the author acted as a reviewer, it was done in the manner already mentioned. The integrated realization actually brings additional problems and limitations, which is fully aware of the desire to offer a completely new platform for emulations, with precise contributions and clearly defined possibilities, which clearly surpasses all previously published solutions. The actual implementation of the offered solution in the form of an integrated circuit is not available to the author due to limitations at his workplace and, most importantly, due to the review deadlines imposed by the journal. Some of the solution configurations, as clearly defined in the last version of the paper, require only one VDCC and very few switches, so that the realization in the form of an integrated circuit becomes very simple. The final dish is of course up to you.

Reviewer 2 Report

I am a little bit perplexed. Without changing the figures with simulation results you changed the description of the used simulation tool from pspice to hspice. You were not aware of the software you use?  I still doubt about your claim "The model used is very detailed, and includes drains/source areas/perimeters" - theese parameters (AS AD PS PD) indeed can be inherited from the .model but are specific for each transistor, dependently on it's aspect ratio. If not - some parasitic capacitances will be over- or under- (usually)estimated. Since you not mentioned any framework/environment like Cadence (usually design kits provide approporiate estimatinions  even at pre-layout phase) I assumed it could be a case. On the other hand - if you used some IC design tool why postlayout simulations were not used for performance estimation? 

Author Response

Reviewer # 2

Comment: I am a little bit perplexed. Without changing the figures with simulation results you changed the description of the used simulation tool from pspice to hspice. You were not aware of the software you use? I still doubt about your claim "The model used is very detailed, and includes drains/source areas/perimeters" - theese parameters (AS AD PS PD) indeed can be inherited from the .model but are specific for each transistor, dependently on it's aspect ratio. If not - some parasitic capacitances will be over- or under- (usually) estimated. Since you not mentioned any framework/environment like Cadence (usually design kits provide approporiate estimatinions even at pre-layout phase) I assumed it could be a case. On the other hand - if you used some IC design tool why postlayout simulations were not used for performance estimation?

Author response: The author must express his displeasure with this attitude of the esteemed reviewer. The author is quite sure which simulation package was used to model the MOS transistors in the simulation procedures. As mentioned in the previous review, in the original version of the paper, the author has misappropriated the models that you can find under HSPICE, which is certainly a major omission by the author, which is explained with enough details in previous evaluation.

Similarly, the authors of all the published papers on this subject have really gone so far, in fact, not to input the details by the esteemed reviewer in their report. I am well aware of the importance of all the comments available and the problems that may arise on this basis. The author is not in a position to look more closely at the platform proposed by the esteemed reviewer because he simply does not have it. On the other hand, there is a clear deadline for responding to comments. The simulation results have not been changed because a model of HSPICE, level 49, was already used in the first version of the work and these are libraries that the author owns. In the last version of the work, the experimental setup was realized in a different operating frequency, which is described and commented in detail in the previous evaluation. The author had to base all his evaluations on the facilities and software that were fully available to him when working and publishing previous papers on the subject. Everyone is free to have the final say in this matter, but I believe that the limits of the author's capabilities must be taken into account. The main idea and desire of the author was to offer a completely new, flexible and, due to the many parameters, superior circuit configuration for emulation of all messages described so far.

In an analysis where the impact of possible process corners (PVT) variations on the circuit operation was studied, the performance was practically tested when the parameters used in the simulation process differed, which may include the impact of the changed dimensions of the MOS transistors used on the parameters found in the HSPICE library. It is practically the standard used in works dealing with the implementation of emulation circuits.

The central contribution of the paper is perfectly well defined and emphasized in several places in the paper, which is fully consistent with the theme of this special issue of the journal. After all the theoretically well-defined achievements, the proposed universal emulator represents an original contribution in the field of memristor element development. The possibilities offered by the proposed configuration include: grounded/floating, incremental/decremental, direct/inverse mode of operation, soft/hard switching; this cannot be found in any of the so far described solutions and structures used to emulate memelements. This also required the complex switching structure described in the paper. Practically, seven completely new and original configurations have been proposed based on VDCC as an active emulation block. Depending on the specific purpose and needs of the future user, the switch structure becomes extremely simple and easy to implement.

Round 3

Reviewer 1 Report

My final decision has been done.

Please read my last my review.